# Assessing the Quality of Oxygen Plasma Focused Ion Beam (O-PFIB) Etching on Polypropylene Surfaces Using Secondary Electron Hyperspectral Imaging

**DOI:** 10.3390/polym15153247

**Published:** 2023-07-30

**Authors:** Nicholas T. H. Farr, Maciej Pasniewski, Alex de Marco

**Affiliations:** 1Department of Materials Science and Engineering, University of Sheffield, Sir Robert Hadfield Building, Mappin Street, Sheffield S1 3JD, UK; 2Insigneo Institute for In Silico Medicine, The Pam Liversidge Building, Mappin Street, Sheffield S10 2TN, UK; 3ExxonMobil Chemical Europe Inc., European Technology Center, 1831 Machelen, Belgium; 4Electron Microscopy for Materials Science (EMAT), University of Antwerp, 2000 Antwerp, Belgium; 5Biomedicine Discovery Institute, Department of Biochemistry and Molecular Biology, Monash University, Melbourne, VIC 3199, Australia; 6Simons Electron Microscopy Center, New York Structural Biology Center, New York, NY 10027, USA

**Keywords:** polymer characterization, secondary electron spectroscopy, surface interactions, focused ion beams, polymer processing

## Abstract

The development of Focused Ion Beam–Scanning Electron Microscopy (FIB-SEM) systems has provided significant advances in the processing and characterization of polymers. A fundamental understanding of ion–sample interactions is still missing despite FIB-SEM being routinely applied in microstructural analyses of polymers. This study applies Secondary Electron Hyperspectral Imaging to reveal oxygen and xenon plasma FIB interactions on the surface of a polymer (in this instance, polypropylene). Secondary Electron Hyperspectral Imaging (SEHI) is a technique housed within the SEM chamber that exhibits multiscale surface sensitivity with a high spatial resolution and the ability to identify carbon bonding present using low beam energies without requiring an Ultra High Vacuum (UHV). SEHI is made possible through the use of through-the-lens detectors (TLDs) to provide a low-pass SE collection of low primary electron beam energies and currents. SE images acquired over the same region of interest from different energy ranges are plotted to produce an SE spectrum. The data provided in this study provide evidence of SEHI’s ability to be a valuable tool in the characterization of polymer surfaces post-PFIB etching, allowing for insights into both tailoring polymer processing FIB parameters and SEHI’s ability to be used to monitor serial FIB polymer surfaces in situ.

## 1. Introduction

The development of Focused Ion Beam–Scanning Electron Microscopy (FIB-SEM) systems has provided significant advances in the processing and characterization of polymers. Innovative developments, including in the operation of multi plasma ion sources, have confirmed FIB-SEM as an essential technique with an ever-increasing number of applications to benefit polymer research [1]. FIB-SEM is now an established technique in materials science, providing polymer microstructural information at a nanometer scale [1]. Oxygen Plasma FIB (O-PFIB) has been developed to create fine polymer cross-sections [2], which can then be imaged by an electron beam or other techniques [3,4,5]. Oxygen beams can produce curtain-free surfaces with minimal polishing, free from undesirable metal implantation. More commonly, O-PFIB-SEM for polymer analysis has been applied in the form of slice and view methods, allowing researchers to generate 3D images faster and at a higher spatial resolution [1,6]. It is well established that FIB etching can vastly change and create unique surface chemistries, but it also needs to be considered that an FIB can affect a specimen’s surface energy. This change in surface energy can result in a fresh reactive surface, which not only attracts unwanted surface contamination [7], but also has the capacity to oxidize surfaces in situ [8]. Electron and ion beam deposition (EBID and IBID) methods are widely employed for a range of applications, for example, EBID has been used to shield surface features prior to ion beam exposure [9]. IBID has been used to create high-aspect-ratio three-dimensional nanostructures with high mechanical strengths [10]. Beam irradiation during EBID and IBID has the potential to contaminate a specimen’s surface through deposition of the chamber’s residual gases. An unknown surface composition post-FIB etching not only creates issues in characterizing and visualizing multiscale structures, but also unknown surface chemistries can strongly affect the mechanical [11] and surface properties of FIB-fabricated structures [7]. 

Despite FIB-SEM being routinely applied in microstructural analyses of polymers, a fundamental understanding of ion–sample interactions is still not fully established. Of particular concern is the lack of opportunity for users to routinely check the chemical composition and consistency of the surfaces produced after an FIB etch. To make progress in this field, in situ analysis of carbon bonding within the SEM chamber (without the need for an Ultra High Vacuum (UHV)) that allows for multiscale scale surface sensitive analysis is required. Additionally, such in situ analysis should be carried out at low beam energies and beam currents to reduce the prospect of sample modification and charging. All these requirements are satisfied via the use of Secondary Electron Hyperspectral Imaging (SEHI) and the accompanying application of Secondary Electron Spectroscopy (SES) [12,13,14]. SEHI is made possible through use of through-the-lens detectors (TLDs) already installed on many available FIB-SEMs. TLDs collect low pass SEs at low primary electron beam energies and currents [15]. The development and application of SEHI has been previously extensively detailed [13,15,16]. In brief, from a collection of SE images taken in the same region of interest, SEHI from different energy ranges can be plotted in addition to SEHI image stacks derived from specific energies. These SEHI stacks can then be processed to compile surface chemistry maps down to the nanoscale [17]. Previous studies have shown SEHI’s ability to characterize a range of polymer systems and detect EBID and IBID contamination within an SEM chamber [8]. For the first time, this study applies SEHI to reveal O–FIB interactions on a polymer’s surface (in this instance, polypropylene). This study not only provides insights into tailoring polymer processing FIB parameters but also shows SEHI’s ability to be used to monitor serial FIB surfaces in situ.

## 2. Materials and Methods

### 2.1. Sample Preparation

Polypropylene (PP) (Achieve™ Advanced PP, provided by ExxonMobil Chemical, Machelen, Belgium) was cryo-faced and coated with 20 nm of gold using a Bal-Tec SCD 050 Sputter Coater (Wetzlar, Germany).

### 2.2. Plasma FIB Exposure

An area of 10 µm × 10 µm was chosen and an FIB trench was created. For the exposure of a O_2_ and Xe^+^ focused ion beam, a Helios UX G4-Hydra system was employed. Initial SEHI data sets were acquired from the PP surface after gold coating. The surface was then exposed to a raster scanning strategy with a 10 µm × 10 µm box pattern using a beam overlap between individual positions of 99% and a pixel dwell time of 500 ns. 

### 2.3. Conventional Low kV Imaging

Surface morphology observations were performed through SEM acquisition at a 1 kV accelerating voltage and 50 pA to avoid sample damage through surface charging [8]. Typical chamber vacuum pressures were in the 10^−6^ mbar range and a working distance of 4 mm was also maintained. For high magnification SE images, a TLD was selected. 

### 2.4. SEHI Data Collection and Processing

The methodology of the application of SEHI has been described in depth previously [8,15]. Briefly, SE spectra were generated using the same acceleration voltage, beam current and working distance as stated for SEM imaging. To mitigate carbon contamination within the SEM chamber, 5 × 25 min plasma chamber cleaning runs were performed prior to data collection. The collection of SE spectra consisting of different energy ranges was enabled by adjusting the mirror electrode voltage (MV) with a tube bias setting of 150 V. Stepping the MV in an energy range of −0.7 to 7 eV was achieved using an automatic iFast collection recipe [18]. Every image was captured at a frame interval of 0.5 s and an MV step size of 0.5 V, which corresponds to a ~0.2 eV electron energy step size. Image processing was undertaken using Fiji ImageJ software (ImageJ2, Version 1.53, open source). 

## 3. Results and Discussion

### 3.1. Capacity of O-PFIB for the Removal of Gold Surface Coatings

Figure 1A presents the SEHI spectra (SES) of cryo-faced and aged PP + gold coating (PPGC) and PP + O_2_ 30 keV 45 nA; the solid line in figure represents the mean profile, while the variation in the SES spectra is highlighted by the shaded area. Previous studies applying SES and SEHI to carbon-based materials have already established that SE peaks within the 2–4.2 eV range are formed from sp^2^/CH_x_ bonding [12,19]. Within the SES of the PPGC, a few minor peaks are visible within the sp^2^ range. However, a dominant peak is present at 6.5 eV. It is considered that this peak is formed in response to the gold coating. As the penetration of SEs at 1 kV is <10 nm and the gold coating is >10 nm, it is expected that SE peaks originating from the underlying PP material would not be attainable due to the high surface sensitivity of SES [15]. Post-PFIB etch (O_2_ 30 KeV 45 nA), there are notable changes within the specimens SES. Most notable is the reduction in the 6.5 eV peak (related to Au emission) and an increase in SE emission within the energy range associated with sp2/CH_x_ bonding. The reduction, but not complete removal, of the Au related SE emission is proposed to be the consequence of three potential scenarios: Firstly, the Au coating has not been fully removed, with trace amounts still identifiable. Secondly, non-etched Au on the edge of the scanning window could generate SEs from the interaction of multiply scattered electrons. Thirdly, as the FIB trench is etched, the Au coating from the edge of trench is redeposited within the walls of the trench. The third scenario is considered to be the most likely.

Nevertheless, this result indicates that post-PFIB etch, the underlying PP is exposed and is identifiable via SEHI/SES. Notably, post-PFIB etch there is a larger variation in the SE spectra compared to that of the original PPGC surface. Figure 1B displays a high-resolution SE image of the PFIB trench produced. The gold coating at the top of the image is clearly visible, as is an area of the underlying and now exposed PP. Figure 1C presents an in-chamber image showing the proximity of the TLD detector, used for SES capture, to the PFIB.

### 3.2. Selection of the Most Appropriate O-PFIB Serial Slicing Parameters

PFIB and FIB systems are routinely used to prepare polymer specimens for SEM or TEM analyses, creating thin slices, cross-sections and polished surfaces. For these examples, the finished material surface must be clean, smooth and undamaged by the FIB process. To better understand the effect O-PFIB has on the surface of PP, three different etch conditions were applied and the resulting SEHI-derived secondary electron spectra (SES) are presented in Figure 2A,B, with accompanying SE images presented in Figure 2C–E.

It can be confirmed that all O-PFIB conditions applied resulted in spectra showing the effective removal of the gold coating from the sample. Observable in Figure 2A is a notable variation in the resulting spectra under different etching conditions. Spectra variation is also notable when the same etching conditions are applied to different regions of the specimen. Figure 1B further indicates this variation within SE peaks is in the range of 3–4.5 eV formed from sp^2^/CHx. The largest variation in spectra collected from different regions is apparent for O_2_ 2 KeV 3.8 nA (conditions commonly applied as a polishing step [20]). It is hypothesized that the increase in chain fragmentation, observed from the reduced accelerating voltage, is a consequence of a highly localized surface effect. The thickness of the amorphization layer formed as a consequence of FIB etching has been theoretically calculated by Monte Carlo simulations to be consistent with that of the penetration of the ion beam [21]. Low kV surface-specific amorphization is therefore not unexpected and can be explained via the restriction of the dissipation of energy through the sample. From the SEHI data provided, this variation indicates polymer chain fragmentation and variation within the surface cross-linking density. Such variation appears consistent with the SE images provided (Figure 2C–E), which show large morphological surface variation for O_2_ 2 KeV 3.8 nA compared to that of other conditions.

### 3.3. Serial O-PFIB Etching

From the data provided in Section 3.2, it would appear that from the conditions analyzed, O_2_ 30 KeV 5.6 nA produces surfaces with the least chemical variation evident within the resulting SEHI spectra. For most polymer PFIB/FIB applications, multiple serial etching is applied. To better understand the consistency of serial etching, O_2_ 30 KeV 5.6 nA was applied and then evaluated using SES spectra. The resulting SEHI spectra are presented in Figure 3A,B. Despite minor peak variations notable within the SE range of 1.4–2.1 eV, previously identified to be related to the molecular order within polymers, the variation in the surface in terms of surface chemistry is consistent after each etch. The resulting SE images (Figure 3C–E) show some minor morphological differences between etches; however, this is inconsistent with any chemical surface changes.

### 3.4. Future Work: Oxygen vs. Xe^+^ for Plasma FIB of PP Specimens

Future studies should aim to apply SEHI to evaluate other polymers and other plasma FIB sources. An example for the scope of this future work is presented in Figure 4A,B. Figure 4A presents the SEHI spectra of PP + O_2_ 30 KeV 5.5 nA and PP + Xe^+^ 30 KeV 4 nA compared to that of the PPGC. PFIB conditions successfully removed the gold coating layer. However, applying Xe^+^ at the conditions selected resulted in a large variation within the SEHI spectra related to cross-linking densities and molecular order. It is expected that the bombardment of Xe^+^ initiates the well-established polyolefin chain fragmentation and recombination reactions [22]. PP free radicals are formed after Xe^+^ collisions, resulting in initial PP chain fragmentation. The creation of free radicals can then trigger a recombination reaction, creating crosslinked PP polymer chains. This result indicates the chain fragmentation action of Xe^+^ is higher than that of oxygen plasma. However, a larger study is required to evaluate various Xe^+^ etching conditions before such a conclusion can confidently be stated. 

## 4. Conclusions

The data presented in this study provide evidence of SEHI’s potential to be a valuable tool in characterizing polymer surfaces post-PFIB etching. SEHI has the capacity to reveal novel insights into a material’s surface chemistry, which can be used to better optimize FIB processing. This study highlights that SEHI can reveal PFIB interactions on the surface of polymers, displaying the capability to monitor serial FIB surfaces in situ. This approach can aid in selecting the most appropriate FIB conditions for the processing of polymers.

## Figures and Tables

**Figure 1 polymers-15-03247-f001:**
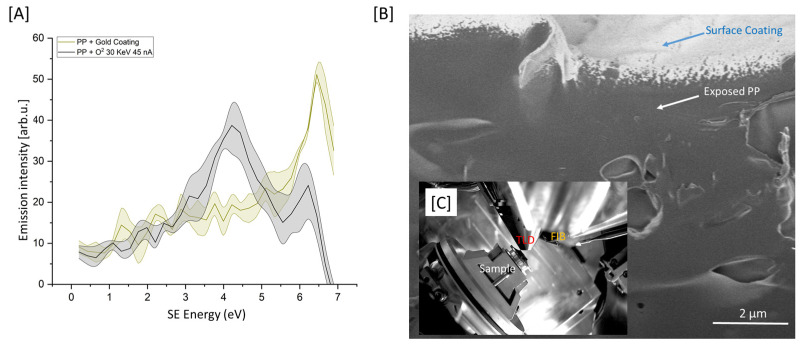
(**A**) SES of the cryo-faced and aged PP + gold coating (*n* = 3) (PPGC) and PP + O_2_ 30 KeV 45 nA (*n* = 3). (**B**) SE image of PP post-PFIB etch. (**C**) In-chamber image displaying the location of the specimen in relation to TLD and O-PFIB.

**Figure 2 polymers-15-03247-f002:**
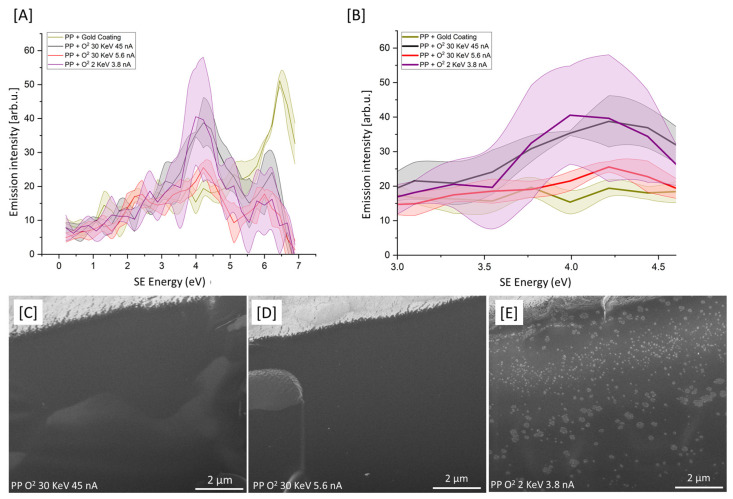
(**A**) SES of cryo-faced and aged PP + gold coating (*n* = 3), PP + O_2_ 30 KeV 45 nA (*n* = 3), PP + O_2_ 30 KeV 5.6 nA (*n* = 3), PP + O_2_ 2 KeV 3.8 nA (*n* = 3). (**B**) SES of samples highlighted in (**A**) plotted from 3 eV to 4.5 eV. (**C**–**E**) SE images of PP post-O-PFIB etch.

**Figure 3 polymers-15-03247-f003:**
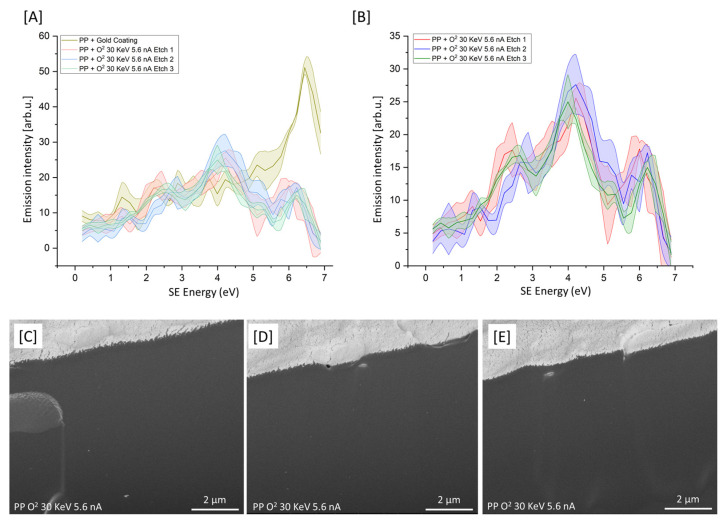
(**A**) SES of cryo-faced and aged PP + gold coating (*n* = 3) and three replicate etches of PP+. (**B**) SES of three replicate etches of O_2_ 30 KeV 5.6 nA (each *n* = 3). (**C**–**E**) SE images of PP post O-PFIB etch.

**Figure 4 polymers-15-03247-f004:**
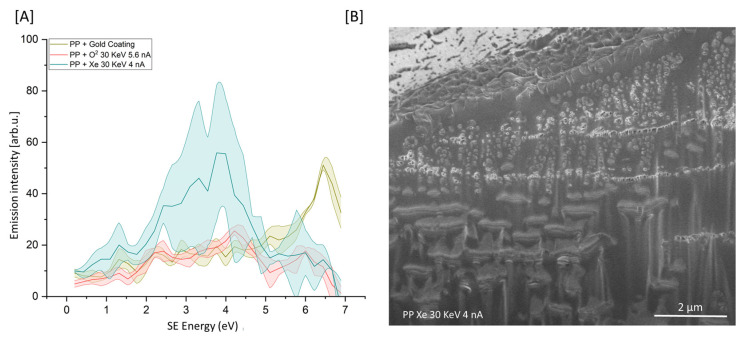
(**A**) SES of cryo-faced and aged PP + gold coating (*n* = 3) (PPGC), PP + O_2_ 30 KeV 5.6 nA (*n* = 3) and PP + Xe^+^ 30 KeV 4 nA (*n* = 3). (**B**) SE image of PP post-Xe-PFIB etch.

## Data Availability

Data available at https://doi.org/10.15131/shef.data.23731686 (24 July 2023).

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
