# Peer review of "Assessing the Quality of Oxygen Plasma Focused Ion Beam (O-PFIB) Etching on Polypropylene Surfaces Using Secondary Electron Hyperspectral Imaging"

_polymers, 2023, doi:10.3390/polym15153247_

Round 1
Reviewer 1 Report
This study applies SEHI to reveal oxygen and xenon plasma FIB interactions on the surface of a polymer. In this Study, it is aiming to use SEHI to realize high spatial resolution and identify carbon bonding present using low beam energies without the needs of UHV. According to the research results presented in the paper, this study has obtained valuable data demonstrating SEHI is capable of characterizing polymer surfaces pose-PFIB etching, which is very important that SEHI could be used to monitor serial FIB polymer surfaces in-situ. Here are the comments:
1. In several sentences, misspelling of “Na” should be corrected as “nA”.
2. Figure 2 -A&B, are a little difficult for readers to tell one curve from the others, please refine the color choices.
3. I would recommend to have some discussion in the conclusions section on the applicable situations as well the exceptions that SEHI could be used to monitor serial FIB surfaces in-situ.
Reviewer 2 Report
General remarks:
The article written by Nicholas T.H. Farr and others presents the application of novel method: secondary electron hyperspectral imaging (SEHI) to assess the quality of focused electron beam induced etching of gold-covered polypropylene. The authors briefly show how to use SEHI as simple, in-situ method, which allows for investigation of areas irradiated with FIB in order to (for example) find the best conditions for ion beam milling. The strength of presented article is that it shows data in a logical order, where one experiment is basing on the results of the previous.
However, I have several remarks to the quality of the research presented here. As far as I understand that this is a communication, which is rather brief form of presentation of scientific results, but in my opinion the article is sometimes missing more detailed analysis of the surface processes involving interactions between ions and polypropylene. In this context the title can be misleading, since the authors do not focus on actual interactions between ions and polypropylene molecules, as the title suggest, but rather show the usage of SEHI as a tool to assess the quality of FIB etching (milling). There are only few, quite general sentences describing the interactions between ions and polymer. The rest of the article rather focuses on very general presentation of the SEHI spectra for different ion energies and beam currents and presenting the difference between them, but without deeper analysis of surface processes. Maybe it will be worth changing the title to one more corresponding with the presented data.
My general impression of the data presentation in this article is that it is a bit sloppy, and it looks like the manuscript was written in a hurry. More than several citations are missing, some sentences are not clear or grammatically incorrect. Please read my detailed comments where I tried to point out these issues.
Despite the negative remarks written above I would still like to give a chance to this study as it contains promising results and suggest re-submission after major revisions.
Detailed comments:
To all spectra presented here: How long does it take to collect one spectrum? Does the deposition of hydrocarbons present in the SEM chamber influence the results of measurement? If yes, how can you minimize this? Please, comment on that.
Line 41: Missing citation in the sentence ending with “… nanometer scale.”
Line 51: FEBID and FIBID have way more applications than what was stated here. Authors did not even listed the most prominent and important examples, especially for the electron beam induced deposition. Please look at this work of Michael Huth, https://doi.org/10.1063/5.0064764 or recent works by Ivo Utke, Harald Plank, Jose Maria de Theresa and others. You can easily find better examples of applications of both EBID and IBID than marked in the text.
Line 66: Citation needed. What about the beam current? The energy of the primary electrons is important factor, indeed, but usage of low energy of electrons (withing the range available at typical SEM) will affect mostly the interaction depth and volume. But also beam intensity will play important role in possible sample modification. Please, comment on that.
Line 88: Sentence is unclear, looks like part of it is missing.
Line 92: Please specify if it was dwell time per pixel or effective dwell time, counting the beam FWHM? Also, what was scanning strategy? Serpentine, rastering? Please, add.
Line 97: Can you please, elaborate on that? Why should this specific working distance prevent charging and damage it causes?
Line 100: Citation(s) needed.
Line 101 -102: This information is basically repetition of what was stated in the paragraph before for the surface morphology observations. Authors can simply state that the acceleration voltage, beam current and working distance during collection of SEHI spectra were the same as for imaging.
Line 118 – 119: There is something wrong with this statement. As the gold coating have less then 10 nm and generated secondary electrons can travel further than 10 nm for 1kV beam, then it would be expected to see the electrons generated in the PP in collected spectrum. Can authors comment on that? Also, where are these values coming from? Simulations? Literature source? If the latter, then there is some citation needed to be add.
Line 122: Word “related” in the brackets is repeated unnecessarily.
Figure 1A: The authors claim that dominant peak at 6.5 eV, visible for the gold coated sample comes from interaction of SE´s with gold layer, and that is why it´s intensity is reduced after the sample was milled with FBI. However, this peak is still present in the spectrum of milled sample. Can authors comment, on that? Is it because the gold layer was not removed completely? Moreover, what is the unit on the vertical axis? (this applies to all spectra presented here).
Figure 1B and C: Please use some markers on the Figure 1B to precisely show the FIB-created trench and the gold layer. Moreover – the authors stated that the FIB irradiated area was 10 x 10 um. The dark area on the SE image, which I assume is the exposed PP is clearly larger. Can you comment on that? Also Figure 1C could be a bit smaller or placed besides 1B, as now it covers more than half of figure 1B.
Line 136, 137: Citation needed.
Figure 2: It is quite subjective, but could you maybe change the colour code of your spectra? Especially for curves for 45 nA and 3.8 nA? They can be challenging to distinguish between.
Line 148: Citation needed for the fragment that applied experimental conditions are the most commonly used for polishing.
Line 149 - 150: Is it possible to estimate what is the range of the variations in cross-linking density on the surface PP for the case of 2kV and 3.8 nA of beam current? Can you also comment why these observed changes are the most visible for these particular experimental conditions?
Figure 3A: Can you explain what is the origin of the brighter region on the surface of PP at the right side of the image? Is it just some morphological change on the surface of polymer?
Line 177: Can you please, elaborate on that statement? Is there any way that you can show how exactly cross linking densities and molecular order can lead to such differences in SEHI spectra?
Line 187: This is quite exaggerated statement basing on the provided data. Please, read my general remarks section.
Please re-read carefully the article as some sentences are not clear or the grammar is incorrect. Please, check my detailed comments.
Round 2
Reviewer 2 Report
The corrected version of the manuscript provided by the authors improved significantly, comparing to the first version I have revised. Especially thanks for the additional explanations of the observed variations in spectra and for corrections of Figures.
After examination of corrected version as well as authors responses to my previous comments I suggest accepting the manuscript with minor corrections listed below.
Line 21: Acronym SEHI should be introduced here.
Line 51 – 52 – the examples, which you use for EBID and IBID are not the most prominent applications for these methods. Please, consider re-phrasing, either adding more appropriate applications or phrases like “among others”.
Line 54: In the context of the introduction, I believe the sentence should rather be phrased that beam irradiation during EBID and IBID can contaminate sample surface by deposition of chamber´s residual gases. Please, clarify this sentence.
Line 103: Please add the citation here. In the cover letter the authors stated that this citation was added, but it is not present in the manuscript.
Line 149 – 150: It is also related to the response to my comment to previous version of the manuscript. As we can see, in presented spectra there is still a small peak visible at the energy of 6.5 eV. I agree with the authors explanation that it can be either matter of re-deposition of gold or some traces of gold still present on the substrate. However, there is still one more possible explanation, which I would like the authors to consider here. While you are scanning to collect SEHI images and SES, if your area of scanning is similar to area of the FIB etch, you can still collect signal coming from the gold bordering your uncovered PP, due to generation of SE´s of the second type: those generated by interaction between backscattered or multiple-times scattered electrons and the sample. These SE´s can be generated in some distance from the beam centre, so if the edge of your SEHI scanning area was close to the gold, this can result in a visible peak in my opinion. Please, comment on that, or add a sentence of explanation if you agree with my explanation. This is relevant not only here, but everywhere you present SEHI spectra after removal of gold.
Line 155: Please add citation here, to the statement in brackets. In the response to my comment the authors stated that the citation was added, but in the manuscript it is still missing. Also instead of “after”, “for” should be used in the fragment at the end of the sentence where experimental conditions for which biggest variations in spectra are observed.
Line 160: “kV” instead of “KV”
Figures: Thanks a lot for increasing the line size, it for sure helped with making the graphs easier to read. One more comment: you can add some line of explanation in the text that the solid line in the graph represents the mean profile, while the range of changes in the SES spectra is visible by the shaded area. I know this is implied in the text but stating it once explicitly would be helpful especially for the younger researchers reading your paper.
No specific comments to the quality of language. Some sentences may be slightly long and complicated, re-read carefully the text and consider re-phrasing some fragments. Please, see my detailed comments.
